# A MET-PTPRK kinase-phosphatase rheostat controls ZNRF3 and Wnt signaling

Minseong Kim[1], Carmen Reinhard[1], Christof Niehrs[1,2]*

[1]Division of Molecular Embryology, DKFZ-ZMBH Alliance, Deutsches Krebsforschungszentrum (DKFZ), Heidelberg, Germany; [2]Institute of Molecular Biology (IMB), Mainz, Germany

**Abstract** Zinc and ring finger 3 (ZNRF3) is a transmembrane E3 ubiquitin ligase that targets Wnt receptors for ubiquitination and lysosomal degradation. Previously, we showed that dephosphorylation of an endocytic tyrosine motif (4Y motif) in ZNRF3 by protein tyrosine phosphatase receptor-type kappa (PTPRK) promotes ZNRF3 internalization and Wnt receptor degradation (Chang et al 2020). However, a responsible protein tyrosine kinase(s) (PTK) phosphorylating the 4Y motif remained elusive. Here we identify the proto-oncogene MET (mesenchymal-epithelial transition factor) as a 4Y kinase. MET binds to ZNRF3 and induces 4Y phosphorylation, stimulated by the MET ligand HGF (hepatocyte growth factor, scatter factor). HGF-MET signaling reduces ZNRF3-dependent Wnt receptor degradation thereby enhancing Wnt/β-catenin signaling. Conversely, depletion or pharmacological inhibition of MET promotes the internalization of ZNRF3 and Wnt receptor degradation. We conclude that HGF-MET signaling phosphorylates- and PTPRK dephosphorylates ZNRF3 to regulate ZNRF3 internalization, functioning as a rheostat for Wnt signaling that may offer novel opportunities for therapeutic intervention.

*For correspondence: niehrs@dkfz.de

**Competing interest:** The authors declare that no competing interests exist.

## Introduction

Wnt signaling plays a pivotal role in embryonic development and in maintaining organ homeostasis (*MacDonald et al., 2009*; *Clevers and Nusse, 2012*; *Kim et al., 2013*). Wnt signaling is tightly regulated at numerous levels of the pathway (*Niehrs, 2012*; *Nusse and Clevers, 2017*; *Anthony et al., 2020*). One prominent mode of regulation occurs at the level of the Wnt receptors LRP6 (low-density lipoprotein receptor-related protein 6) and FZD (Frizzled class receptors), which are antagonized by various secreted factors and activity-modulated by phosphorylation of their cytoplasmic tail (*Niehrs and Shen, 2010*; *Cruciat and Niehrs, 2013*). In addition, Wnt receptor availability at the plasma membrane is controlled by ER chaperones (*Hsieh et al., 2003*; *Weekes et al., 2012*; *Berger et al., 2017*) as well as by the E3 ubiquitin ligase ZNRF3 and its functional homolog RNF43. These transmembrane proteins act as negative feedback regulators of Wnt signaling that promote receptor degradation by ubiquitylating and internalizing cell surface LRP6 and FZD (*Hao et al., 2012*; *Koo et al., 2012*). More recently, ZNRF3 was shown to also internalize and degrade the BMP receptor type 1 A and thereby antagonize BMP signaling (*Lee et al., 2020*). ZNRF3/RNF43 are counteracted by secreted proteins of the R-spondin family, which sequester ZNRF3/RNF43 in conjunction with LGR4/5/6, thereby leading to the membrane clearance of ZNRF3/RNF43 (*Carmon et al., 2011*; *de Lau et al., 2011*; *Glinka et al., 2011*; *Hao et al., 2012*; *Koo et al., 2012*).

ZNRF3 and RNF43 play important roles in adult stem cells and in tumorigenesis (*de Lau et al., 2014*; *Hao et al., 2016*). Yet, little is known about how ZNRF3 and RNF43 are regulated intracellularly (*Jiang et al., 2015*; *Ci et al., 2018*; *Giebel et al., 2021*). We recently discovered an endocytic signal in

the cytoplasmic domain of ZNRF3 consisting of four consecutive tyrosine residues (4Y motif: **Y**ETM**Y**-QH**YY**) that regulates ZNRF3- and in turn Wnt receptor internalization (*Chang et al., 2020*). Tyrosine motifs play a critical role in regulating endocytosis of transmembrane proteins and unphosphorylated YXXX$\phi$ and $\phi$XXY sites as well as YXX$\phi$ ($\varphi$ = bulky hydrophobic amino acid) serve as internalization motifs (*Zhang and Allison, 1997*; *Roush et al., 1998*; *Bonifacino and Traub, 2003*; *Royle et al., 2005*). Phosphorylation of the 4Y motif prevents ZNRF3 internalization and stabilizes Wnt receptors at the cell surface. Conversely, dephosphorylation of the 4Y motif by PTPRK (protein tyrosine phosphatase receptor-type kappa) promotes ZNRF3 internalization and Wnt receptor clearance from the cell surface, thereby inhibiting Wnt signaling (*Figure 1A*). While our report showed that 4Y phosphorylation regulates ZNRF3 trafficking, it left unanswered the crucial question of what is the protein tyrosine kinase(s) (PTK) responsible for 4Y phosphorylation.

Here we identify the proto-oncogene MET (mesenchymal-epithelial transition factor/hepatocyte growth factor receptor) as a 4Y kinase. MET is a single transmembrane receptor PTK with important roles in promoting cell growth and differentiation during embryonic development, organogenesis, and cancer progression (*Trusolino et al., 2010*; *Gherardi et al., 2012*; *Bradley et al., 2017*). The only known ligand of MET is hepatocyte growth factor/scatter factor (HGF/SF). HGF binding to MET mediates dimerization and enzymatic activation of MET to engage multiple downstream effectors such as AKT, MAPK, STAT, and SRC (*Trusolino et al., 2010*; *Malik et al., 2020*). HGF-MET signaling also cross-talks with Wnt signaling pathway at various levels (*Papkoff and Aikawa, 1998*; *Monga et al., 2002*; *Apte et al., 2006*; *Huang et al., 2012*; *Koraishy et al., 2014*; *Chaudhary et al., 2019*). Here we show that HGF stabilizes binding of MET to ZNRF3 and induces tyrosine phosphorylation of the 4Y motif. HGF-MET signaling attenuates endocytosis of ZNRF3 and thereby stabilizes Wnt receptors at the cell surface, leading to β-catenin activation. Our results establish an HGF-gated rheostat whereby phosphorylation/dephosphorylation by MET and PTPRK tunes ZNRF3 endocytic trafficking and Wnt receptor availability.

## Results
### MET is a ZNRF3 4Y kinase

To identify PTKs responsible for phosphorylating the 4Y motif of ZNRF3, we consulted the kinome database GPS 3.0 (*Xue et al., 2011*) for computational prediction of phosphorylation sites with their cognate protein kinases, which revealed a number of putative 4Y tyrosine kinases (*Figure 1B*). To this list, we added some additional PTKs that either display transmembrane (RTKs) or cortical localization. To screen these 18 PTKs, we utilized H1703 human lung adenocarcinoma cells harboring doxycycline (Dox)-inducible ZNRF3-HA (TetOn ZNRF3-HA) to overcome poor transfection efficiency of this cell line as well as the general lack of antibodies against endogenous ZNRF3 (*Chang et al., 2020*). We monitored tyrosine phosphorylation of ZNRF3 by immunoblot using a pan phospho-Tyr (pTyr) antibody upon treatment with a selected library of 40 chemical inhibitors targeting the 18 PTKs. To increase sensitivity of pTyr-ZNRF3 detection, here and in other experiments monitoring specifically pTyr-ZNRF3, we employed bafilomycin to prevent lysosomal degradation and stabilize the protein (*Chang et al., 2020*). Among 18 selected candidate kinases, inhibition of SRC family kinases (SFKs), focal adhesion kinase (FAK; encoded by PTK2), and MET showed significant reduction on pTyr-ZNRF3 (*Figure 1C*). To validate this finding, we monitored pTyr-ZNRF3 upon overexpression of SRC, FAK, and MET in HEK293T cells (*Figure 1D*). Overexpression of MET strongly- and SRC weakly phosphorylated ZNRF3 (lanes 5–6), while FAK (lane 7) had no effect. SRC, YES, and FYN are widely expressed SFKs with overlapping functions (*Stein et al., 1994*; *Thomas et al., 1995*), but we counted-out YES as candidate since a YES-specific inhibitor had no effect in the initial screen. We therefore continued with individual or combined knockdown of SRC, FYN, and MET by siRNAs and found that depletion of all three kinases, most prominently of MET, reduced ZNRF3 phosphorylation (*Figure 2A*; compare lane 2 to lanes 3, 4, or 6). Triple knockdown of SRC, FYN, and MET nearly eliminated the pTyr signal to background (lanes 7 vs. 1). Of note, while MET is often amplified in cancer cells, H1703 cells harbor normal MET copy number (*Kubo et al., 2009*).

Since SFKs are downstream effectors of MET (*Herynk et al., 2007*; *Bertotti et al., 2010*), we wondered whether the effect of MET knockdown on pTyr-ZNRF3 was indirectly caused by reduced SFK activity. However, the activity of SFKs as monitored by pY416 autophosphorylation was unchanged

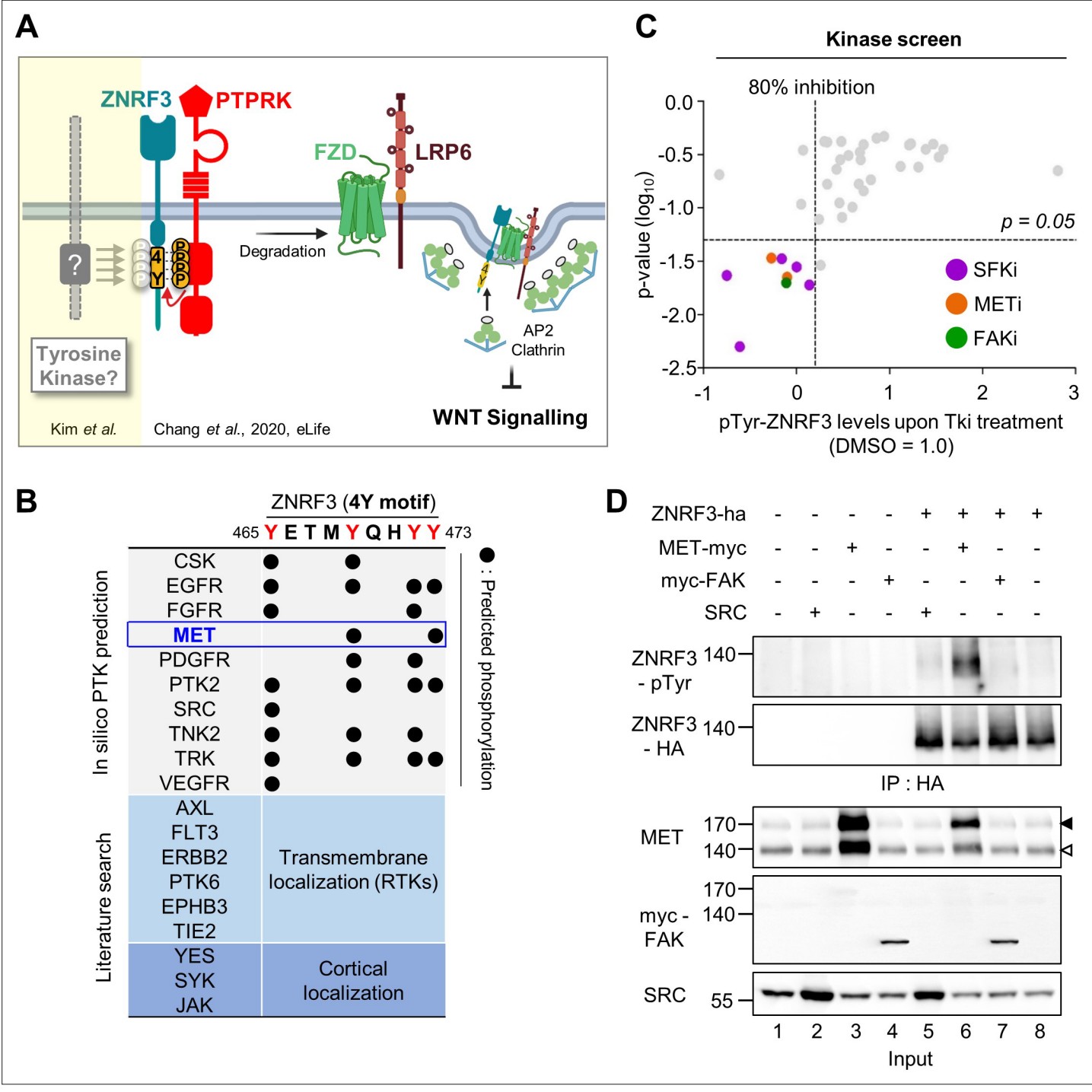

**Figure 1.** Identification of SRC and MET as ZNRF3 kinases. (**A**) Model for tyrosine phosphorylation regulating ZNRF3 and Wnt signaling (*Chang et al., 2020*). ZNRF3 E3 ubiquitin ligase reaching the plasma membrane is co-internalized with Wnt receptors and targets them for lysosomal degradation to reduce Wnt signaling. An unphosphorylated 4-tyrosine (4Y) motif serves as ZNRF3 internalization signal. The phosphatase PTPRK dephosphorylates and unmasks the 4Y motif, promoting internalization and lysosomal targeting of ZNRF3 and Wnt receptors, and reducing Wnt signaling. PTPRK is counteracted by an unknown tyrosine kinase(s) that phosphorylates the 4Y motif, impairs ZNRF3/Wnt receptor internalization, and increases Wnt signaling. Created with Biorender.com. (**B**) Scheme of 4Y PTK candidate selection. Eighteen tyrosine kinases were selected by combined in silico prediction and reported cellular localization. Kinase inhibitor screening (**C**) was then conducted for validation. (**C**) Inhibitor screen for ZNRF3 4Y kinases. TetOn ZNRF3-HA H1703 cells were treated with 40 inhibitors targeting 18 PTKs selected as shown in (**B**). pTyr-ZNRF3 was analyzed by immunoblot for pan-phosphotyrosine and normalized to total ZNRF3. Results from two independent screens were pooled to generate the dot graph. X-axis indicates the relative tyrosine phosphorylation of ZNRF3 upon inhibitor (Tki) treatment relative to DMSO control, y-axis indicates p-value. Three kinases (SFK,

*Figure 1 continued on next page*

*Figure 1 continued*

MET, FAK) with phosphorylation inhibition ≥80 % and p-value ≤ 0.05 were selected for downstream analysis and are highlighted. (**D**) Analysis of pTyr-ZNRF3 in HEK293T cells co-transfected with ZNRF3-HA and MET-myc, myc-FAK or SRC. Cell lysates were pulled down with anti-HA antibody and subjected to immunoblot analysis using the indicated antibodies. Data show a representative result from three independent experiments with similar outcome. Black arrowhead: Immature MET precursor. White arrowhead: Mature, processed MET (β-chain). Source files of all blots used in this figure are available in *Figure 1—source data 1*.

The online version of this article includes the following source data for figure 1:

**Source data 1.** Uncropped immunoblot images for *Figure 1D*.

upon knockdown of MET (*Figure 2A*), suggesting that MET directly regulates pTyr-ZNRF3 independently of SFKs.

Reduced pTyr-ZNRF3 caused by MET depletion was confirmed by pulldown of ZNRF3 (*Figure 2B–C*). Moreover, ZNRF3 robustly bound to endogenous MET, whereas no binding to SRC was detected (*Figure 2D*). To test whether SRC or MET indeed mediates tyrosine phosphorylation of the 4Y motif, we transfected them along with the wild-type (Wt) ZNRF3 or mutant ZNRF3(Δ4Y) lacking the 4Y motif and monitored pTyr-ZNRF3 levels (*Figure 2E*). ZNRF3 phosphorylation by MET was dramatically reduced with ZNRF3(Δ4Y), although not completely abolished, suggesting some 4Y-independent phosphorylation by MET. In contrast, no reduction but a strong increase was observed for SRC-mediated ZNRF3 phosphorylation (*Figure 2E*). This unexpected increase may be explained by our previous observation that ZNRF3(Δ4Y) preferentially localizes at the cell surface (*Chang et al., 2020*) where activated SRC also resides.

We previously showed that depletion of PTPRK induces 4Y phosphorylation of ZNRF3 by opposing the action of an unknown tyrosine kinase (*Chang et al., 2020*). To address whether PTPRK opposes MET in this context, we carried out 'rescue' experiments with the competitive MET kinase inhibitor Crizotinib. Indeed, enhanced pTyr-ZNRF3 induced by PTPRK siRNA depletion was rescued by Crizotinib (*Figure 2F*). This PTPRK-MET double-inhibition experiment supports that MET and PTPRK oppose each other in ZNRF3 4Y phosphorylation (*Figure 2G*).

Taken together, our results show that MET binds ZNRF3 and is both necessary and sufficient for 4Y phosphorylation, supporting the notion that MET is indeed a 4Y kinase that opposes PTPRK. Of note, MET is likely not the only 4Y kinase since in early *Xenopus* embryos, ZNRF3 dephosphorylation by PTPRK plays a role during Spemann organizer patterning (*Chang et al., 2020*) before the onset of significant Met expression (*Koibuchi et al., 2004*) and hence we did not analyze Met in *Xenopus*. SRC on the other hand promotes ZNRF3 phosphorylation likely at a site different from the 4Y motif and was not further analyzed. In the following analysis, we therefore focused on the MET-ZNRF3 interaction.

## HGF-MET signaling triggers 4Y phosphorylation of ZNRF3

Given that HGF ligand binding to MET initiates its kinase activity, we tested if HGF-MET signaling stimulates ZNRF3 phosphorylation. We treated TetOn ZNRF3-HA H1703 cells with human HGF and analyzed pTyr-ZNRF3 after pulldown. Indeed, HGF strongly induced pTyr-ZNRF3, and this induction was reduced by the MET inhibitor Crizotinib (*Figure 3A*), as well as by siRNA-mediated knockdown of MET (*Figure 3B*). Moreover, Co-IP analysis showed that HGF treatment increased the binding of ZNRF3 to endogenous MET in a time-dependent fashion, peaking at 30 min of HGF treatment (*Figure 3C*). Increased binding of ZNRF3 to MET occurred despite total MET levels slightly decreasing after 15 min HGF treatment, the latter likely reflecting ligand-induced MET internalization and degradation (*Taher et al., 2002*). Furthermore, while HGF strongly induced phosphorylation of Wt ZNRF3, phosphorylation of ZNRF3(Δ4Y) was barely affected (*Figure 3D*), corroborating that HGF-MET signaling mainly targets the 4Y motif. We conclude that HGF-MET signaling induces 4Y phosphorylation of ZNRF3 (*Figure 3E*).

## HGF-MET signalling stabilizes ZNRF3 at the cell surface

We previously showed that the unphosphorylated 4Y motif is an endocytic signal for ZNRF3. Depletion of PTPRK increases ZNRF3 levels at the cell surface because phosphatase inhibition allows phosphorylation to accumulate and mask the endocytic motif (*Chang et al., 2020*). This finding raised the possibility that HGF-MET signaling stabilizes cell surface ZNRF3, while MET inhibition leads to ZNRF3 internalization. To test these predictions we performed cell surface biotinylation assays and monitored

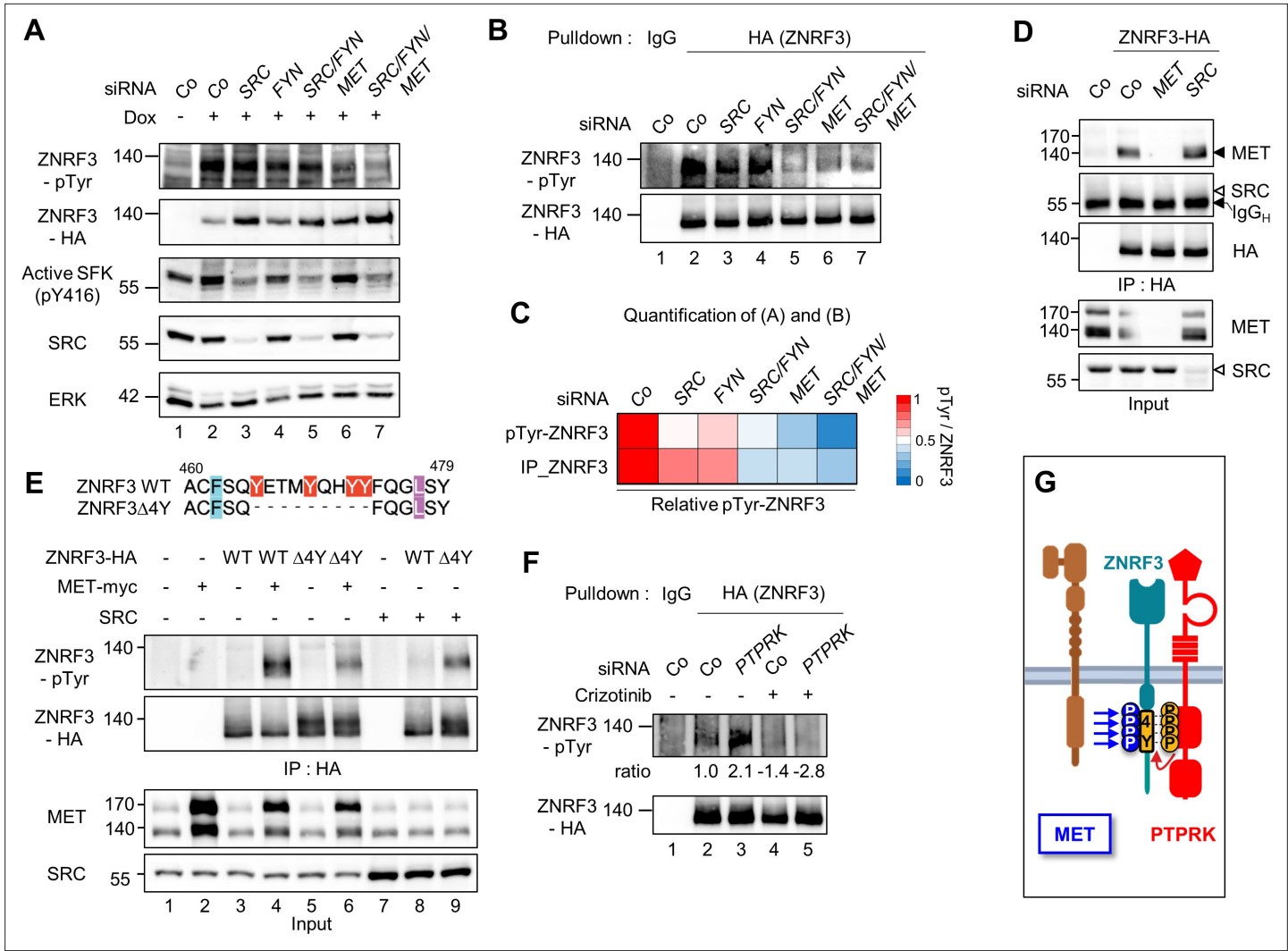

**Figure 2.** MET is a ZNRF3 4Y kinase. (**A**) Tyrosine phosphorylation of ZNRF3 requires endogenous MET and SRC. pTyr-ZNRF3 was monitored in TetOn ZNRF3-HA H1703 cells upon siRNA knockdown of *SRC, FYN, MET* or combinations as indicated. Cells were induced with Dox for 48 hr before harvest. Cells without Dox treatment were used as control for background pTyr signal. Data show a representative result from two independent experiments with similar outcome. (**B**) Analysis of pTyr-ZNRF3 in TetOn ZNRF3-HA H1703 cells upon siRNA knockdown of *SRC, FYN, MET* or combination as indicated. Lysates were pulled down with anti-HA antibody or control IgG and subjected to immunoblot analysis with the indicated antibodies. (**C**) Visualization of normalized pTyr-ZNRF3 from immunoblots (**A**) and ZNRF3 pulldown (**B**) as a heat map. Background signal (lane 1 from (**A, B**)) was subtracted from pTyr-ZNRF3 before normalization to total ZNRF3. Color code is represented on the right. (**D**) ZNRF3 binds endogenous MET but not SRC. Co-immunoprecipitation (Co-IP) analysis in TetOn ZNRF3-HA H1703 cells upon siRNA knockdown of *MET* or *SRC*. Data show a representative result from three independent experiments with similar outcome. Black arrow: IgG heavy chain. Black arrowhead: MET. White arrowhead: expected position of SRC. (**E**) MET but not SRC is a 4Y-specific kinase. Tyrosine phosphorylation of ZNRF3-HA or ZNRF3(Δ4Y)-HA in HEK293T cells co-transfected with the indicated kinases. Lysates were pulled down with anti-HA antibody and subjected to immunoblot analysis using the indicated antibodies. Data show a representative result from three independent experiments with similar outcome. (**F**) MET and PTPRK oppose each other on 4Y phosphorylation. Analysis of pTyr-ZNRF3 in TetOn ZNRF3-HA H1703 cells upon siRNA knockdown of *PTPRK* and/or overnight crizotinib treatment as indicated. Lysates were pulled down with anti-HA antibody or control IgG and subjected to immunoblot analysis with the indicated antibodies. Ratio, relative pTyr-ZNRF3 normalized to total ZNRF3. Data show a representative result from two independent experiments with similar outcome. (**G**) Model: MET and PTPRK phosphorylate and dephosphorylate the 4Y motif on ZNRF3, respectively. Created with Biorender.com. Source files of all blots used in this figure are available in *Figure 2—source data 1*. Source files of densitometric analysis for heat map in (**C**) are available in *Figure 2—source data 2*. A source file of densitometric analysis for (**F**) is available in *Figure 2—source data 3*.

The online version of this article includes the following source data for figure 2:

**Source data 1.** Uncropped immunoblot images for *Figure 2A–B, D–F*.

**Source data 2.** Densitometric analysis of immunoblots for *Figure 2A–B* to generate heap map for *Figure 2C*.

**Source data 3.** Densitometric analysis of immunoblots for *Figure 2F*.

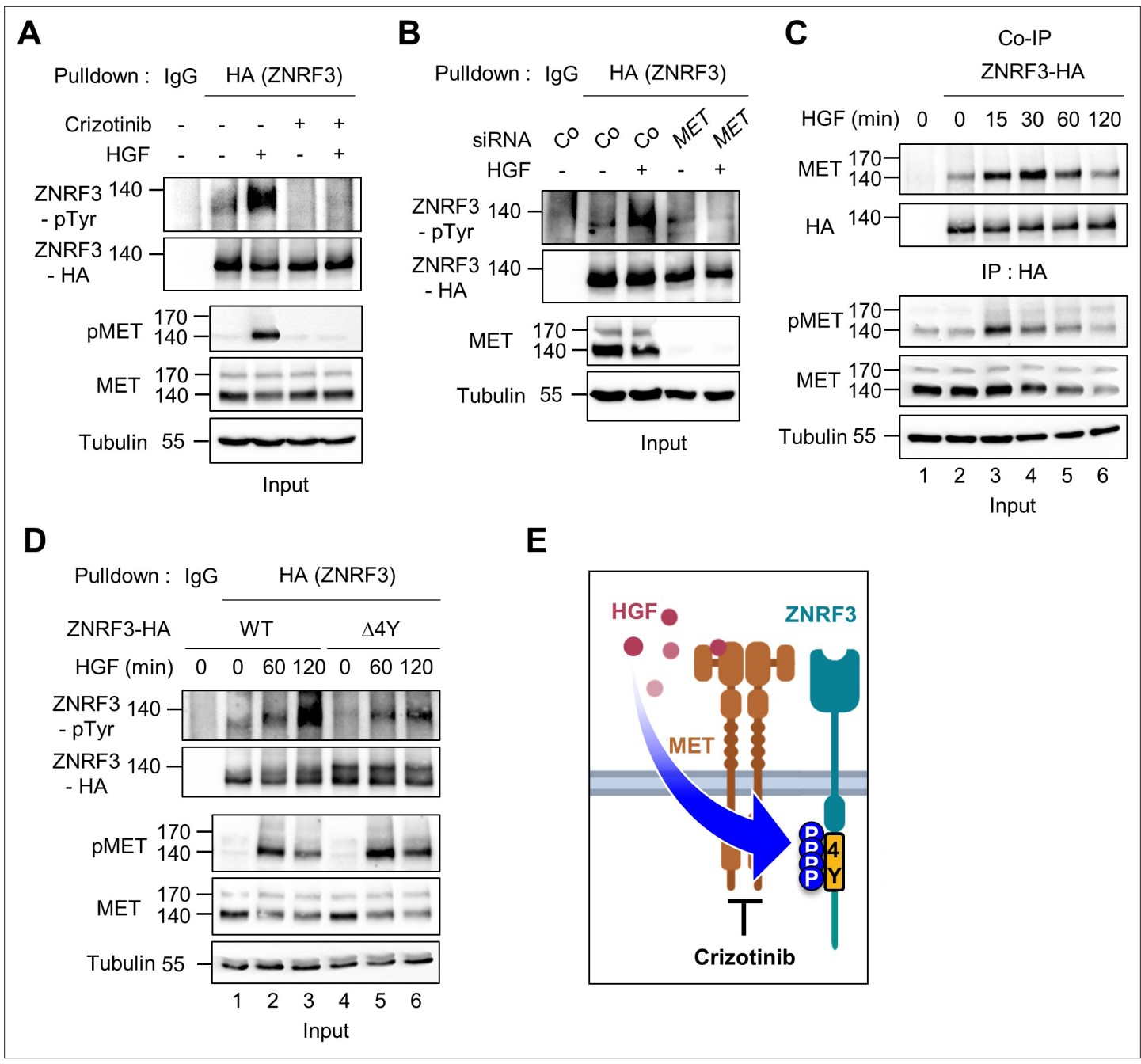

**Figure 3.** HGF-MET signaling triggers 4Y phosphorylation of ZNRF3. (**A**) HGF induces tyrosine phosphorylation of ZNRF3 and requires MET kinase activity. Analysis of pTyr-ZNRF3 in TetOn ZNRF3-HA H1703 cells upon overnight Crizotinib treatment. HGF was added for 2 hr before harvest. Lysates were pulled down with anti-HA antibody or control IgG and subjected to immunoblot analysis. Data show a representative result from two independent experiments with similar outcome. (**B**) HGF-induced tyrosine phosphorylation of ZNRF3 requires MET. pTyr-ZNRF3 in TetOn ZNRF3-HA H1703 cells was analyzed upon siRNA knockdown of *MET*. Cells were treated with HGF for 2 hr before harvest. Lysates were pulled down with anti-HA antibody or control IgG and subjected to immunoblot analysis using the indicated antibodies. Data show a representative result from two independent experiments with similar outcome. (**C**) HGF promotes the interaction of endogenous MET with ZNRF3. Co-IP analysis in TetOn ZNRF3-HA H1703 cells upon HGF treatment. Lysates were pulled down anti-HA antibody or control IgG and subjected to immunoblot analysis. Data show a representative result from two independent experiments with similar outcome. (**D**) HGF promotes tyrosine phosphorylation of the 4Y motif. Analysis of tyrosine phosphorylation of ZNRF3-HA (WT) or ZNRF3(Δ4Y)-HA in H1703 cells upon HGF treatment. Lysates were pulled down with anti-HA antibody or control IgG and subjected to immunoblot analysis. Data show a representative result from two independent experiments with similar outcome. (**E**) Model: HGF-MET phosphorylates 4Y motif on ZNRF3. Crizotinib blocks the 4Y phosphorylation by HGF. Created with Biorender.com. Source files of all blots used in this figure are available in *Figure 3—source data 1*.

*Figure 3 continued on next page*

*Figure 3 continued*

The online version of this article includes the following source data for figure 3:

**Source data 1.** Uncropped immunoblot images for *Figure 3A–D*.

ZNRF3. To antagonize MET signaling, we employed Crizotinib and MET siRNA, both of which reduced cell surface levels of ZNRF (*Figure 4A–B*). In contrast, neither the FAK inhibitor TAE226 nor the SFK inhibitor Dasatinib changed cell surface ZNRF3 levels (*Figure 4—figure supplement 1A, B*). Conversely, HGF treatment increased ZNRF3 surface levels in a time-dependent manner, plateauing after ~45 min (*Figure 4C*). We conclude that HGF-MET signaling stabilizes ZNRF3 at the cell surface consistent with a role in phosphorylating and masking the endocytic 4Y motif (*Figure 4D–E*).

## HGF-MET signaling stabilizes Wnt receptors by counteracting ZNRF3 function

Endocytosis of ZNRF3 driven via the 4Y motif is required to induce Wnt receptor internalization and degradation (*Chang et al., 2020*). Hence, 4Y phosphorylation by HGF-MET signaling may serve to counteract Wnt receptor clearance. To analyze if HGF prevents ZNRF3-mediated FZD degradation, we co-transfected FZD5 and ZNRF3 and monitored FZD5 upon HGF treatment in H1703 cells (*Figure 5A*). FZD5 shows a lower band on immunoblot and a set of upper bands, known to correspond to immature and mature receptor protein, respectively. Overexpression of ZNRF3 reduced mature FZD5 levels as previously reported (*Hao et al., 2012*; *Koo et al., 2012*; *Giebel et al., 2021*) (compare lanes 1 and 2). HGF reversed this reduction after 90 min treatment (lanes 7–8). We tested if the ability of HGF to stabilize FZD5 against ZNRF3 depends on the 4Y motif. Strikingly, HGF treatment rescued reduced FZD5 levels in Wt ZNRF3- but hardly in ZNRF3(Δ4Y) transfected cells (*Figure 5B*).

To test if HGF-MET signaling also stabilizes the Wnt coreceptor LRP6, we analyzed total LRP6 by immunoblot, which also shows both upper and lower bands, known to correspond to mature post-Golgi and immature LRP6 forms, respectively. HGF specifically increased mature LRP6 levels within 1 hr treatment, and this increase by HGF was drastically abolished by si*ZNRF3/RNF43* treatment, which constitutively increased LRP6 levels (*Figure 5C*). We tested an alternative lung adenocarcinoma cell line, H3122, and inhibiting MET with Crizotinib reduced mature LRP6, an effect that was almost abolished when ZNRF3/RNF43 where siRNA-inhibited (*Figure 5—figure supplement 1A*). Of note, despite the fact that it does not contain any obvious endocytic Tyr motifs, we also knocked down RNF43 in the above experiments, because ZNRF3 and RNF43 can heterodimerize (*Yu et al., 2020*). Taken together the results support that MET signaling is necessary and sufficient for LRP6 regulation via ZNRF3 and in multiple lung cancer cell lines.

Finally, we analyzed if HGF treatment also induces downstream Wnt/β-catenin signaling by monitoring β-catenin accumulation upon Wnt3a treatment. Consistently, combined HGF and Wnt3a treatment synergized to induce β-catenin accumulation (*Figure 5D–E*). Moreover, the HGF effect was MET- and ZNRF3/RNF43 dependent (*Figure 5—figure supplement 1B, C*), confirming that HGF-MET promotes Wnt signaling via ZNRF3/RNF43.

Our findings suggest that HGF resembles the role of R-spondin in Wnt signaling, which also regulate ZNRF3 endocytosis, albeit by a different mechanism (*Hao et al., 2016*). We therefore asked if HGF and RSPO3 cooperate in Wnt signaling. We treated cells with a sub-optimal dose of Wnt3a and added HGF or RSPO3 alone or in combination and then monitored β-catenin accumulation. Interestingly, HGF and RSPO3 strongly synergized in inducing β-catenin accumulation (*Figure 5—figure supplement 1D*), suggesting that they act non-redundantly.

Taken together, these results support a model whereby the ZNRF3 4Y motif is a phospho-regulated molecular rheostat of endocytosis: HGF-MET signaling impairs ZNRF3 function via phosphorylating and masking its endocytic 4Y motif, leading to accumulation of Wnt receptors and enhanced Wnt signalling, which is opposed by PTPRK (*Figure 5F*).

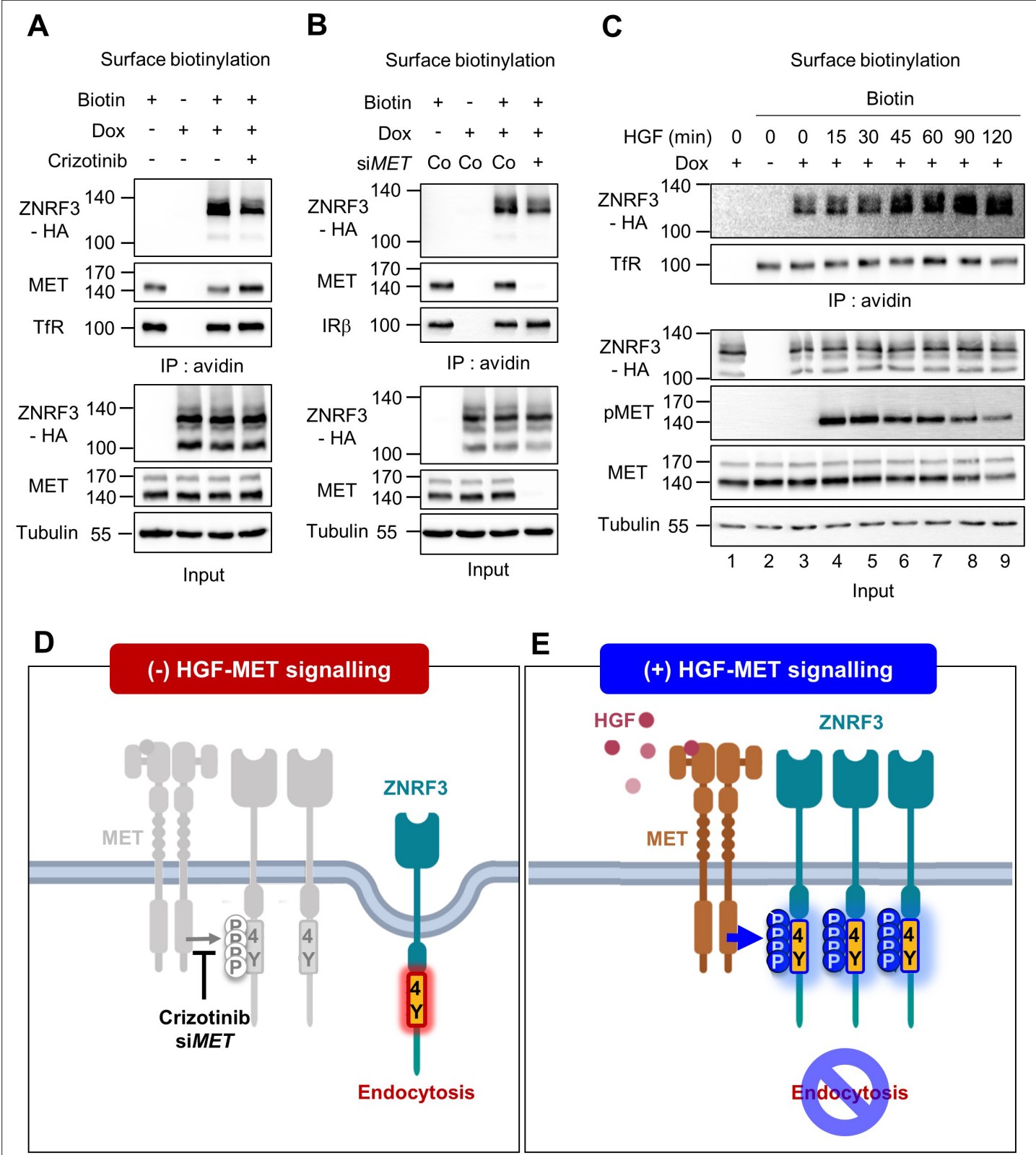

**Figure 4.** HGF-MET signaling stabilizes ZNRF3 at the cell surface. (**A–B**) Cell surface biotinylation assay of ZNRF3 in TetOn ZNRF3-HA H1703 cells upon Crizotinib treatment (**A**) or siRNA knockdown of *MET* (**B**). Cells were treated with Crizotinib (**A**) overnight before harvest. After labeling surface proteins with biotin, lysates were pulled down with streptavidin beads and subjected to immunoblot analysis. Transferrin receptor (TfR) or Insulin receptor beta (IRβ) was used as a loading control for avidin pulldown. Tubulin was used as a loading control for total cell lysate (Input). Data show a representative

*Figure 4 continued on next page*

*Figure 4 continued*

result from two independent experiments with similar outcome. (**C**) HGF stabilizes cell surface ZNRF3 within 45 min. Cell surface biotinylation assay in TetOn ZNRF3-HA H1703 cells upon HGF treatment for the indicated time. After labeling surface protein with biotin, lysates were pulled down with streptavidin beads and subjected to immunoblot analysis. Transferrin receptor (TfR) was used as loading control for avidin pulldown. Tubulin was used as loading control for total cell lysate (Input). Data show a representative result from two independent experiments with similar outcome. (**D–E**) Model: In the absence of HGF-MET signaling (**D**), unphosphorylated 4Y motif promotes endocytosis of ZNRF3. In the presence of HGF-MET signaling (**E**), activated MET phosphorylates and masks the endocytic 4Y motif, stabilizing ZNRF3 at the cell surface. Created with Biorender.com. Source files of all blots used in this figure are available in *Figure 4—source data 1*.

The online version of this article includes the following source data and figure supplement(s) for figure 4:

**Source data 1.** Uncropped immunoblot images for *Figure 4A–C*.

**Figure supplement 1.** Surface ZNRF3 is not changed by FAK or SFKs inhibitor.

**Figure supplement 1—source data 1.** Uncropped immunoblot images for *Figure 4—figure supplement 1A–B*.

## Discussion

### MET phosphorylation tunes ZNRF3 trafficking and function

Through in silico analysis combined with kinase inhibitor screening we identified the proto-oncogene MET as a receptor tyrosine kinase that phosphorylates the endocytic 4Y motif of ZNRF3. We further show that HGF induces the MET-ZNRF3 interaction and tyrosine phosphorylation of the 4Y motif, thereby attenuating ZNRF3 endocytosis and Wnt receptor degradation. Together with our previous report that PTPRK is a 4Y phosphatase (*Chang et al., 2020*), the results argue for a model whereby phosphorylation/dephosphorylation of its 4Y endocytic motif functions as a rheostat that tunes ZNRF3 function and Wnt signalling at the receptor level (*Figure 5F*).

Several lines of evidence support that MET directly phosphorylates ZNRF3: the phosphorylated 4Y motif conforms to a MET consensus phosphorylation site, MET is both necessary and sufficient for ZNRF3 phosphorylation, HGF stimulates ZNRF3 phosphorylation, and MET binds and phosphorylates ZNRF3 in a HGF-dependent manner. The fact that HGF stimulates ZNRF3 binding suggests that MET dimerization or autophosphorylation is important for their interaction. Indeed, MET is known to complex with a remarkably diverse set of other transmembrane proteins, notably cell adhesion molecules and other RTKs, thereby acting as a hub for signal integration and amplification (*Crepaldi et al., 1997*; *Follenzi et al., 2000*; *Jo et al., 2000*; *Trusolino et al., 2001*; *Giordano et al., 2002*; *Orian-Rousseau et al., 2002*; *Tanizaki et al., 2011*).

It is likely that MET is not the only 4Y kinase since in early *Xenopus* embryos, ZNRF3 dephosphorylation by PTPRK plays a role in Spemann organizer patterning at a stage that precedes significant zygotic expression of *Xenopus* Met (*Koibuchi et al., 2004*). Additional 4Y kinases besides MET may also explain while we failed to observe a role for MET in HEK29T cells (data not shown). While we found that SRC is not a 4Y kinase, it nevertheless phosphorylates ZNRF3 and our preliminary data suggest that it may inhibit Wnt signaling (data not shown).

Autophosphorylation of the MET kinase domain at Tyr1234/1235 is critical for kinase activation and in vitro, a Tyr1235 phosphopeptide is dephosphorylated by several PTPRK-like phosphatases (*Barr et al., 2009*). Moreover, in HEK293 cells PTPRJ/DEP-1 dephosphorylates MET phospho-sites required for adapter protein binding (*Palka et al., 2003*). Since both MET and PTPRK bind ZNRF3, it will be interesting to explore if all three proteins form a ternary complex in which MET kinase-PTPRK phosphatase cross regulate each other.

Regarding HGF vs. R-spondin function toward ZNRF3, both are secreted proteins that antagonize this E3 ubiquitin ligase by regulating its endocytosis. However, they do so in an opposite manner: R-spondins bind ZNRF3 in conjunction with LGR4 and co-internalize the complex to remove ZNRF3, thereby de-repressing Wnt receptors (*Carmon et al., 2011*; *de Lau et al., 2011*; *Glinka et al., 2011*; *Hao et al., 2012*; *Koo et al., 2012*). HGF does the opposite: it prevents ZNRF3 co-internalization of Wnt receptors by inducing ZNRF3-MET association and 4Y phosphorylation. How can either promoting or impairing endocytosis both inhibit ZNRF3? This apparent paradox may be due to HGF-MET and R-spondins targeting either distinct ZNRF3 surface complexes or different endocytic routes. HGF and RSPO3 strongly synergized in inducing β-catenin accumulation. Relatedly, since MET binds ZNRF3, it will be interesting to test if R-spondins might co-internalize and reduce MET levels at the cell surface. The observation that HGF and RSPO3 strongly synergize in inducing β-catenin accumulation

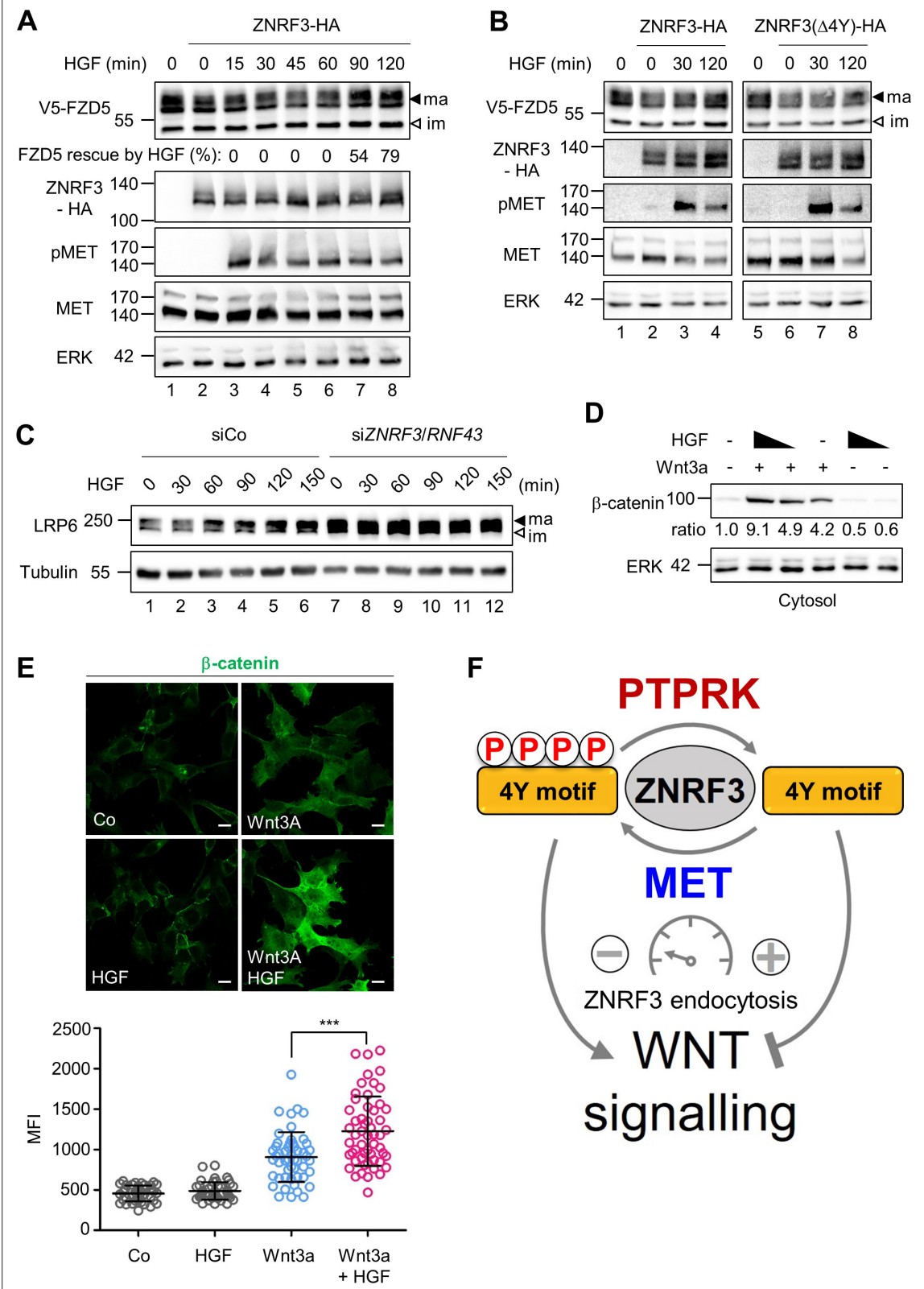

**Figure 5.** HGF increases Wnt receptor levels by counteracting ZNRF3 function. (**A**) HGF protects mature FZD5 against ZNRF3. Immunoblot analysis of V5-FZD5 in H1703 cells with ZNRF3-HA transfection upon HGF treatment as indicated. ma, im: Mature and immature forms of V5-FZD5, respectively. Rescue of ZNRF3-reduced FZD5 levels by HGF was quantified by normalizing V5-FZD5 to ERK, setting V5-FZD5 in lanes 1 and 2 to 100- and 0%, respectively, and calculating HGF-rescued levels of V5-FZD5 relative to this scale. Data show a representative result from two independent experiments

*Figure 5 continued on next page*

*Figure 5 continued*

with similar outcome. (**B**) HGF stabilizes mature FZD5 in presence of ZNRF3 but not ZNRF3(Δ4Y). Immunoblot analysis of V5-FZD5 in H1703 cells with ZNRF3-HA or ZNRF3(Δ4Y)-HA transfection upon HGF treatment as indicated. ma, im: Mature and immature forms of V5-FZD5, respectively. Data show a representative result from two independent experiments with similar outcome. (**C**) HGF stabilizes LRP6 levels in a ZNRF3/RNF43-dependent manner. Immunoblot of total LRP6 protein in H1703 cells upon HGF treatment without or with si*ZNRF3/RNF43* knockdown as indicated. ma, im: Mature and immature form of LRP6, respectively. (**D**) HGF enhances β-catenin levels upon Wnt3a stimulation. Immunoblot analysis of cytosolic (saponin-extracted) β-catenin in H1703 cells treated overnight as indicated. Ratio, relative levels of β-catenin normalized to ERK1/2. Data show a representative result from three independent experiments with similar outcome. (**E**) HGF enhances β-catenin levels upon Wnt3a stimulation. Immunofluorescence microscopy (IF) showing nuclear and cytosolic β-catenin in H1703 cells. Cells were treated overnight as indicated. Top, representative IF images. Bottom, quantification of β-catenin (Mean ± SD, ***$p<0.001$, student t-test, MFI: Mean fluorescence intensity). (**F**) Model for how the phospho-regulated 4Y motif acts as a molecular rheostat for ZNRF3 endocytosis in Wnt signaling. MET phosphorylates and PTPRK dephosphorylates the 4Y motif, which attenuates or promotes ZNRF3 endocytosis to activate or inhibit Wnt signaling, respectively. Source files of all blots used in this figure are available in *Figure 5—source data 1*. A source file of densitometric analysis for Figure 5 (**A and D**) is available in *Figure 5—source data 2*.

The online version of this article includes the following source data and figure supplement(s) for figure 5:

**Source data 1.** Uncropped immunoblot images for *Figure 5A–D*.

**Source data 2.** Densitometric analysis of immunoblots for *Figure 5A and D*.

**Figure supplement 1.** RSPO and HGF synergize to increase Wnt signaling.

**Figure supplement 1—source data 1.** Uncropped immunoblot images for *Figure 5—figure supplement 1A–D*.

**Figure supplement 1—source data 2.** Densitometric analysis of immunoblots for *Figure 5—figure supplement 1A–D*.

supports that they target distinct ZNRF3 pools and raises the possibility that HGF may be useful in growing Wnt-dependent organoids.

## Cross-talk between MET and Wnt signalling

We show that the net effect of HGF-MET signaling is to stabilize Wnt receptors and β-catenin. This conclusion is in keeping with other observations that MET activates Wnt signaling by signal cross-talk at entry points distinct from ZNRF3. HGF-MET signaling in renal cells stimulates GSK3-mediated LRP6 phosphorylation to promote Wnt signaling (*Monga et al., 2002*; *Koraishy et al., 2014*). HGF-AKT signaling was shown to decrease GSK3 activity and increase β-catenin levels (*Papkoff and Aikawa, 1998*; *Ishibe et al., 2006*). MET can also associate with- and phosphorylate β-catenin, promoting its nuclear translocation and Wnt signaling (*Monga et al., 2002*; *Apte et al., 2006*). Furthermore, HGF signaling up-regulates Wnt ligands, integrins, and LEF1 to control downstream Wnt target gene activation (*Huang et al., 2012*; *Chaudhary et al., 2019*). However, in H1703 cells such downstream-acting mechanisms that bypass Wnts may not be involved since without Wnt3a, HGF did not induce β-catenin accumulation and because HGF-enhanced Wnt signaling was abolished by si*ZNRF3/RNF43* treatment (*Figure 5D–E* and *Figure 5—figure supplement 1C*).

ZNRF3 is a general inhibitor of Wnt receptors of both the LRP6 and FZD family (*Hao et al., 2016*). In addition, ZNRF3 was recently shown to inhibit BMPR1A-BMP4 signaling (*Lee et al., 2020*). This suggests that MET inhibiting ZNRF3 may not only upregulate canonical Wnt signaling but also cross-talk with other FZD-dependent pathways such as Wnt/STOP (*Taelman et al., 2010*; *Acebron et al., 2014*), Wnt/PCP, and Wnt/Ca2+ (*Miller et al., 1999*; *Hao et al., 2012*; *Tsukiyama et al., 2015*), as well as BMPR1A-BMP4 signaling. The reported HGF-induced hyperactivation of BMP-SMAD signaling would be consistent with this possibility (*Addante et al., 2020*).

The cross-talk between MET and Wnt signaling is cell-biologically intriguing because of their shared property to stimulate cell proliferation, anti-apoptosis, angiogenesis, and invasive behavior (*Breuhahn et al., 2006*; *Gonzalez and Medici, 2014*). For example, both pathways induce a master regulator of invasion, *snail* (*Saint-Jeannet et al., 1997*; *Grotegut et al., 2006*). MET- and Wnt pathway activation are also both prominently involved in colorectal cancer (CRC) progression (*Bradley et al., 2017*; *Schatoff et al., 2017*). While the majority of Wnt-pathway-dependent tumors are caused by mutations in APC (adenomatous polyposis coli) that lead to ligand-receptor independent signaling, there is a class of tumors that requires Wnt-receptor signaling and displays distinct molecular pathogenesis, morphology, and prognosis (*Kleeman and Leedham, 2020*). Our results suggest that therapy of this subclass of tumors may benefit from combined MET and Wnt inhibitor treatment by promoting ZNRF3 trafficking and Wnt receptor clearance.

# Materials and methods

**Key resources table**

| Reagent type (species) or resource | Designation | Source or reference | Identifiers | Additional information |
|---|---|---|---|---|
| Gene (*Homo sapiens*) | MET | NCBI | NM_000245 | |
| Gene (*Homo sapiens*) | SRC | Gary Davidson PMID: 25391905 | | |
| Gene (*Homo sapiens*) | PTK2 | GenBank | AAH35404.1 | |
| Cell line (*Homo-sapiens*) | H1703 | ATCC | CRL-5889 RRID:CVCL_1490 | |
| Cell line (*Homo-sapiens*) | H3122 | Cellosaurus | RRID:CVCL_5160 | Prof. Dr. Rocio Sotillo |
| Cell line (*Homo-sapiens*) | 293T | ATCC | CRL-3216 RRID:CVCL_0063 | |
| Cell line (*Homo-sapiens*) | H1703 TetOn ZNRF3-HA | *Chang et al., 2020* | | Generated from H1703 |
| Antibody | Anti-MET (Rabbit monoclonal) | Cell signaling | 8198T RRID:AB_10858224 | WB (1:3000) |
| Antibody | Anti-phospho-MET (Y1234/1235; Rabbit monoclonal) | Cell signaling | 3077T RRID:AB_2143884 | WB (1:1000) |
| Antibody | Anti-α-Tubulin (Mouse monoclonal) | Sigma | T5168 RRID:AB_477579 | WB (1:3000) |
| Antibody | Anti-SRC (Rabbit monoclonal) | Cell signaling | 2,123 S RRID:AB_2106047 | WB (1:1000) |
| Antibody | Anti-phospho Src Family (Y416; Rabbit monoclonal) | Cell signaling | 2,101 S RRID:AB_331697 | WB (1:1000) |
| Antibody | Anti-Insulin receptor beta (Rabbit monoclonal) | Cell signaling | 3025T RRID:AB_2280448 | WB (1:1000) |
| Antibody | Anti-GAPDH (Rabbit monoclonal) | Cell signaling | 2,118 L RRID:AB_561053 | WB (1:10000) |
| Antibody | Anti-β-catenin (mouse monoclonal) | BD bioscience | 610154 RRID:AB_397555 | WB (1:5000) IF (1:500) |
| Antibody | Goat anti-mouse Alexa 488 (goat polyclonal) | Invitrogen | A11029 RRID:AB_138404 | IF (1:500) |
| Transfected construct (human) | siRNA to human MET (SMARTpool) | Horizon discovery | M-003156-02-0005 | 25 nM |
| Transfected construct (human) | siRNA to human ZNRF3 (SMARTpool) | Horizon discovery *Chang et al., 2020* | | |
| Transfected construct (human) | siRNA to human RNF43 (SMARTpool) | Horizon discovery *Chang et al., 2020* | | |
| Recombinant DNA reagent | pDONR223-MET | Addgene | Kit #1000000014 Plasmid #23,889 | |
| Recombinant DNA reagent | pDONR223-PTK2 | Addgene | Kit #1000000014 Plasmid #23,902 | |
| Recombinant DNA reagent | pDEST-Myc-C-term | DKFZ Vector repository | MYC-C | Source: Stefan Pusch, University Heidelberg |

*Continued on next page*

*Continued*

| Reagent type (species) or resource | Designation | Source or reference | Identifiers | Additional information |
|---|---|---|---|---|
| Recombinant DNA reagent | pDEST-Myc-N-term | DKFZ Vector repository | MYC-N | Source: Stefan Pusch, University Heidelberg |
| Recombinant DNA reagent | pDEST-hSRC | Gary Davidson PMID: 25391905 | | |
| Peptide, recombinant protein | Recombinant human HGF | Peprotech | 100–39 H | 50 ng/ml |
| Chemical compound, drug | Kinase Screening Library (96well) | Cayman chemical | 10505 | 500 nM |
| Chemical compound, drug | Crizotinib | Cayman chemical | 12087 | 500 nM |
| Chemical compound, drug | Dasatinib | Cayman chemical | 11498 | 500 nM |
| Chemical compound, drug | TAE226 | Cayman chemical | 17685 | 500 nM |

## Cell culture

H1703 cells (ATCC) and H3122 (Cellosaurus) were maintained in RPMI with 10 % FBS, supplemented with 2 mM L-glutamine, 1 mM sodium pyruvate and penicillin/streptomycin. HEK293T cells (ATCC) were maintained in DMEM with 10 % FBS, supplemented with 2 ml L-glutamine and 1 mM penicillin/streptomycin. Cell identity was authenticated in ATCC based on STR profiling. Regular mycoplasma tests confirmed that both cell lines were negative for mycoplasma contamination.

## Cell transfection and treatment

For measuring pTyr-ZNRF3 upon transient DNA transfection, HEK293T or H1703 cells were grown to ~40 % confluency and transfected using X-tremeGENE9 (Roche, Basel, Switzerland) or Lipofectamine 3000 (Invitrogen, Carlsbad, CA), respectively. After 24 hr, cells were treated with 20 nM bafilomycin (Calbiochem, San Diego, CA) for 24 hr before harvest. For monitoring V5-FZD5, H1703 cells were transfected using Lipofectamine 3000 for 48 hr then treated with 50 ng/ml recombinant human HGF (Peprotech, Rocky Hill, NJ) as indicated. For measuring pTyr-ZNRF3 in TetOn ZNRF3-HA H1703 cells, cells were transfected with 25 nM of each siRNA using DharmaFECT1 (Horizon discovery, Cambridge, UK) reagent. After 24 hr, 200 ng/ml Doxycycline was added to induce ZNRF3 expression. On the following day, cells were treated with 20 nM bafilomycin for 24 hr before harvest. Where indicated, 100 ng/ml recombinant human HGF was added for 2 hr before harvest unless indicated otherwise. For monitoring MET binding to ZNRF3 using TetOn ZNRF3-HA H1703 cells, ZNRF3 was induced for 48 hr with 200 ng/ml doxycycline, and then cells were treated with 100 ng/ml recombinant human HGF as indicated. To monitor LRP6 level in H1703 or H3122 cells, 48 hr after siRNA transfection, 100 ng/ml human HGF was treated as indicated or 100 nM Crizotinib was added overnight before harvest. To measure cytosolic β-catenin, 20 % Wnt3a conditioned media and/or 100 ng/ml human HGF were added overnight before harvest. In *Figure 5—figure supplement 1D*, 5% Wnt3a, 1 % RSPO3 conditioned media and/or 100 ng/ml human HGF were added 2 hr before harvest.

## Expression constructs

MET construct tagged with MYC at the C-terminus (pDEST-MET-MYC) was generated by Gateway cloning by inserting MET ORF (Open reading frame) from pDonr223-MET (Addgene; #23889) into pDEST-MYC(C-term) (DKFZ vector repository) FAK construct tagged with MYC at the N-terminus (pDEST-MYC-FAK)was generated analogously using FAK ORF from pDonr223-PTK2 (Addgene; #23902) and pDEST-MYC(N-term) (DKFZ vector repository). Constructs were validated by sequencing. pDEST-hSRC was kindly provided by Gary Davidson (*Chen et al., 2014*).

## Immunoblot and immunoprecipitation

To isolate total cell lysates, cells were harvested in cold PBS and lysed with Triton lysis buffer (20 mM Tris-Cl, pH 7.5, 1 % Triton X-100, 150 mM NaCl, 1 mM EDTA, 1 mM EGTA, 1 mM β-glycerophosphate, 2.5 mM sodium pyrophosphate, 1 mM Na-orthovanadate) supplemented with complete protease inhibitor cocktail (Roche, Basel, Switzerland). To obtain cytosolic fractions, cells were lysed with Saponin lysis buffer (20 mM Tris-Cl, pH 7.5, 0.05 % Saponin, 1 mM $MgCl_2$, 1 mM Na-orthovanadate) supplemented with complete protease inhibitor cocktail (Roche, Basel, Switzerland). Lysates were cleared by centrifugation, and a Bradford assay was performed to measure the protein concentration. For immunoblot, 30 µg of lysates were mixed with NuPage LDS sample buffer containing 50 mM DTT and heated at 70 °C for 10 min. For *Figure 5A–B*, samples were not heated to prevent aggregation of FZD5 protein.

For co-immunoprecipitation or pull-down assays, 300–800 µg of total cell lysates were precleared with 10 µl of Protein A/G Plus Agarose (Santacruz Biotechnologies, Santacruz, CA) on a rotator at 4 °C for 1 hr. Precleared lysates were incubated with 20 µl of A/G Plus Agarose with anti-HA (1867423; Roche, Basel, Switzerland) on a rotator at 4 °C for 4 hr. Immunoprecipitated proteins were washed with Triton lysis buffer four times and mixed with NuPage LDS sample buffer containing 50 mM DTT, followed by heating at 70 °C for 10 min. Samples were subjected to SDS-PAGE, transferred to nitro-cellulose membrane, and blocked with 5 % BSA in TBST (10 mM Tris-Cl, pH 8.0, 150 mM NaCl, 0.05 % Tween-20). Primary antibodies in blocking buffer were applied overnight at 4 °C, and incubation of secondary antibodies was carried out at RT for 1 hr. Immunoblots were developed with SuperSignal West pico ECL (Thermo Scientific, Waltham, MA) and analyzed using a LAS-3000 (Fujifilm, Tokyo, Japan). Densitometry analyses were done with Multi-gauge software (Fujifilm, Tokyo, Japan). Antibody information is listed in key resource table.

## Kinase inhibitor screening

Human ZNRF3 was analyzed with the kinase prediction online tool, GPS 3.0 (http://gps.biocuckoo.cn/index.php, Wuhan, China) to select candidates for 4Y kinase screening. Additional PTK candidates were selected based on plasma membrane or cortical localization (*Shuai and Liu, 2003*; *Ingley, 2008*; *Mocsai et al., 2010*). For screening, TetOn ZNRF3-HA H1703 cells were induced with doxycycline (DOX) (200 ng/ml). After 24 hr, tyrosine kinase inhibitors (500 nM; stocks in DMSO) from a Kinase screening library (Cayman, Ann Arbor, MI) were added together with bafilomycin (20 nM) for additional 24 hr before harvest. DMSO was used as a control. Cells were lysed with Triton lysis buffer (20 mM Tris-Cl, pH 7.5, 1 % Triton X-100, 150 mM NaCl, 1 mM EDTA, 1 mM EGTA, 1 mM β-glycero-phosphate, 2,5 mM sodium pyrophosphate, 1 mM Na- orthovanadate) supplemented with complete protease inhibitor cocktail (Roche, Basel, Switzerland), followed by centrifugation and samples were subjected to SDS-PAGE analysis and immunoblot for pan-phosphotyrosine (pTyr-ZNRF3), HA (total ZNRF3) and α-tubulin. For the quantitative analysis, background signals (cells without Dox treatment) were subtracted from phospho-tyrosine signals before normalizing to total ZNRF3. Dot graph was generated with Graphpad (Prism) using average pTyr signal and p-value (student t-test).

## Cell surface biotinylation assay

TetOn ZNRF3-HA H1703 cells were induced with doxycycline (200 ng/ml). Where indicated, cells were transfected with siRNA for 72 hr or treated with chemical inhibitors (500 nM) for 24 hr and washed three times with cold PBS. Surface proteins were biotinylated with 0.25 mg/ml sulfo-NHS-LC-LC-Biotin (Thermo scientific, Waltham, MA) at 4 °C for 30 min. For non-biotinylated control, PBS was added. The reaction was quenched by washing 3 times with 10 mM monoethanolamine and cells were harvested and lysed with Triton lysis buffer. 200–300 µg of lysate was incubated with 10 µl streptavidin agarose (Thermo scientific, Waltham, MA) to pull-down biotinylated surface proteins, and precipitated proteins were subjected to immunoblot analysis as indicated.

## Immunofluorescence microscopy

Cells were grown on coverslips in 12-well plates, followed by Wnt3a and/or HGF treatment overnight. Cells were fixed in 4 % PFA for 10 min, permeabilized and blocked (PBS, 0.1 % Tween, 4 % normal goat serum) for 1 hr. β-catenin antibodies (1:500) were incubated overnight at 4 °C, and goat-anti

mouse Alexa 488 (1:500) was applied for 1 hr at room temperature. Coverslips were mounted with Fluoromount-G. Images were obtained using LSM 700 (Zeiss). Quantification was done using Image J.

## Acknowledgements

We thank F Cong (Novartis) for providing the ZNRF3 constructs; G Davidson for providing the SRC construct and R Sotillo (DKFZ) for providing H3122 cells. We thank expert technical support by the DKFZ core facility for light microscopy. We thank A Hirth for help with BioRender and thank H Lee for illustrations. This work was funded by the Deutsche Forschungsgemein-schaft (DFG, German Research Foundation) – SFB 1324/2, B01.

## Additional information

### Funding

| Funder | Grant reference number | Author |
| --- | --- | --- |
| Deutsche Forschungsgemeinschaft | SFB 1324/2, B01 | Minseong Kim<br>Carmen Reinhard<br>Christof Niehrs |

The funders had no role in study design, data collection and interpretation, or the decision to submit the work for publication.

### Author contributions

Minseong Kim, Conceptualization, Investigation, Validation, Visualization, Writing - original draft, Writing - review and editing; Carmen Reinhard, Resources, Validation; Christof Niehrs, Conceptualization, Funding acquisition, Project administration, Supervision, Writing - original draft, Writing - review and editing

### Author ORCIDs

Minseong Kim http://orcid.org/0000-0002-3927-4899
Christof Niehrs http://orcid.org/0000-0002-9561-9302

### Decision letter and Author response

Decision letter https://doi.org/10.7554/eLife.70885.sa1
Author response https://doi.org/10.7554/eLife.70885.sa2

## Additional files

### Supplementary files

• Transparent reporting form

• Supplementary file 1. Data of Inhibitor screen for ZNRF3 4Y kinases. Table of inhibitor screen results. The table includes the name of inhibitors, relative pTyr-ZNRF3 from two screens, average, and p-value.

### Data availability

All data generated or analyzed during this study are included in the manuscript and supporting file; Source Data files have been provided for every figure.

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
