## [Decision Letter]

**Acceptance summary:**

The study presents a novel mode of crosstalk between hepatocyte growth factor (HGF)-MET and Wnt signaling. The receptor tyrosine kinase MET phosphorylates an endocytic motif in the cytosolic tail of the E3 ligase ZNRF3, which is implicated in Wnt signaling. As a result, phosphorylated ZNRF3 is stabilized at the membrane and is prevented from down-regulating Wnt receptors, a process involving endocytosis. The work has implications for both HGF-MET and Wnt signaling. In a broader sense, the work forms a good example of interactions between signaling pathways leading to new ways of regulation.

**Decision letter after peer review:**

Thank you for submitting your article "A MET-PTPRK kinase-phosphatase rheostat controls ZNRF3 and Wnt signalling" for consideration by *eLife*. Your article has been reviewed by 2 peer reviewers, including Roel Nusse as Reviewing Editor and Reviewer #1, and the evaluation has been overseen by Michael Eisen as the Senior Editor.

As you will see, the reviewers are overall interested in the new data and would put value on the work as it will lead to further understanding of the Wnt pathway. The paper does indeed complement your earlier publication on the role of ZNRF3 phosphorylation.

However, there is one major concern, something that you no doubt are aware of, and that the use of bafilomycin to prevent lysosomal degradation and stabilize proteins. Both reviewers ask for clarification and what would happen in the absence of bafilomycin.

Reviewer 2 wonders how general the findings are, as the experiments were performed in a single cell line (H1703).

Both reviewers raise several other issues that are probably easier to address in a revised version, which we hope you will be able to submit.

*Reviewer #1:*

Kim at al., identify the proto-oncogene MET as a tyrosine kinase on ZNRF3, phosphorylating the 4Y motif which is in itself involved in internalization of the ZNRF3 receptor. They show that MET binds to ZNRF3 and induces 4Y phosphorylation, and is stimulated by the MET ligand HGF. This is important for the understanding of the mechanisms of Wnt signaling, an important developmental signaling pathway. The work is of high quality, with extensive experimental tests and appropriate controls.

Figure 1 B shows is a diagram with different classes of TM kinases, including a light blue block of "transmembrane tyrosine kinases". But the grey block above also includes some transmembrane tyrosine kinases. What is the logic of subdivision of these kinases?

On page 5, it is stated that bafilomycin was added to prevent lysosomal degradation and stabilize the protein. But some experiments, in Figure 4 for example, test whether HGF-MET signalling stabilizes ZNRF3 at the cell surface. What happens to those stabilizing data in the absence of bafilomycin?

Figure 3 C shows binding data between ZNRF3 and MET, before and after HGF addition to cells. There seems to be a differences between the amount of pMET input (bottom panel), total MET and MET brought down by ZNRF3-HA, comparing 15 and 30 minutes. Perhaps the authors can comment on this.

The figures are nicely done, with the diagrams. But the yellow zone in Figure 1G is not explained.

*Reviewer #2:*

A previous publication of the Niehrs group identified a regulatory role of the phosphatase PTPRK in promoting ZNRF3-mediated turnover of Wnt receptors (Chang et al., *eLife* 2020). The current study presents a follow-up of this work in which the authors search for the kinase responsible for phosphorylation of the 4Y endocytic motif within the ZNRF3 cytosolic tail. They identify and validate the MET receptor as a kinase of the ZNRF3-4Y motif by using ZNRF3-HA-inducible H1703 cells in combination with siRNAs depletion or overexpression of putative kinases.

In line, overexpressed ZNRF3 was found in complex with endogenous MET. The role of MET is further confirmed by an HGF-induced increase in ZNRF3 phosphorylation, and decreased 4Y phosphorylation and ZRNF3 internalization upon treatment with Crizotinib, a MET inhibitor. Moreover, HGF treatment was shown to increase the levels of ZNRF3 at the cell surface and, at the same time, prevent ZNRF3-mediated surface removal of the Wnt receptors FZD5 and LRP6.

Overall, this study adds valuable insights in how the Wnt pathway negative feedback regulator ZNRF3 is regulated by phosphorylation and how these events depend on crosstalk between the Wnt and HGF pathways. Limitations of this study include a lack of support for the broad and biological relevance of the proposed mechanism, and the reliance on the use of bafilomycin treatment in all experiments, which may confound protein trafficking behavior. In addition, the relative contribution of crosstalk between Wnt and HGF pathways at the level of ZNRF3 regulation remains unclear.

1. All experiments revealing a functional link between MET and ZNRF3 were performed in a single cell line (H1703 cells), and findings in this cell line could not be reproduced in HEK293T cells (discussion). Also, a lack of functional/biological readouts prevents an interpretation of the biological relevance of the presented model. A demonstration that the presented mechanism is conserved and broadly relevant to other cells and model systems would strengthen this study.

2. The authors state that all experiments were performed in the presence of Bafilomycin A1 (p5), which has many caveats, in particular when evaluating protein trafficking behavior. Can the authors exclude that their findings are a merely a curiosity of conditions in which endosomal acidification is prevented? Based on the literature, endosomal organization, endocytosis as well as recycling rates are affected by these conditions. To substantiate the proposed model, at least some of the functional experiments (e.g. HGF treatment-mediated alterations in β-catenin levels, ZNRF3 cell surface stabilization, FZD and LRP6 turnover assays) should be performed in absence of the drug.

3. For HGF and Wnt pathways, multiple points of crosstalk have been reported. To judge the relative contribution of HGF-MET regulation of ZNRF3 versus previously reported crosstalk events, the levels of HGF-induced β-catenin stabilization and β-catenin-dependent reporter activity should be evaluated in ZNRF3/RNF43-ko cells.

4. The authors make a comparison between the activities HGF and RSPO. Both factors mediate pathway activation by acting on ZRNF3, albeit by different mechanisms that involve either ZNRF3 stabilization at the plasma membrane (HGF) or an increase in ZNRF3 turnover (RSPO). The outcome of both pathways is Wnt receptor stabilization. To judge the relative contribution of HGF-mediated pathway stimulation, can the authors make a direct comparison of RSPO and HGF-mediated potentiation of Wnt-induced pathway activation, using reporter assays? Also, do these pathways act synergistically?

5. In line 234-236, the authors speculate that HGF inhibition might be beneficial for treatment of WNT-dependent cancers. However, as WNT-dependent colorectal cancers commonly display a lack of ZNRF3 expression, either due to mutation or gene silencing (e.g. Lannagan et al., Gut 2019), this poses a conceptual problem.

---

## [Author Response]

Reviewer #1:Kim at al., identify the proto-oncogene MET as a tyrosine kinase on ZNRF3, phosphorylating the 4Y motif which is in itself involved in internalization of the ZNRF3 receptor. They show that MET binds to ZNRF3 and induces 4Y phosphorylation, and is stimulated by the MET ligand HGF. This is important for the understanding of the mechanisms of Wnt signaling, an important developmental signaling pathway. The work is of high quality, with extensive experimental tests and appropriate controls.Figure 1 B shows is a diagram with different classes of TM kinases, including a light blue block of "transmembrane tyrosine kinases". But the grey block above also includes some transmembrane tyrosine kinases. What is the logic of subdivision of these kinases?

The grey block represents PTKs from in silico tyrosine kinase prediction. In addition, we have manually added several transmembrane or cortical localized TKs from literature search (blue and light blue blocks). We realized that this misunderstanding is due to inaccurate labeling for the diagram and have clarified this point.

On page 5, it is stated that bafilomycin was added to prevent lysosomal degradation and stabilize the protein. But some experiments, in Figure 4 for example, test whether HGF-MET signalling stabilizes ZNRF3 at the cell surface. What happens to those stabilizing data in the absence of bafilomycin?

We apologize for the misunderstanding regarding bafilomycin treatment (p5). Bafilomycin was only used in the few western blot analyses that analyzed pTyr-ZNRF3, because the tyrosine phosphorylation signal is very weak and labile. Bafilomycin was not added in any other experiments. The bafilomycin trick to boost pTyr-ZNRF3 was already employed in Chang et al., where we demonstrated qualitatively the same- but weaker effects without bafilomycin (see Author response image 1) . We now have modified the sentence.

**Author response image 1. sa2fig1:** Bafilomycin treatment enhances pTyr-ZNRF3 levels. Tyrosine phosphorylation of ZNRF3 in TetOn ZNRF3-HA H1703 cells upon siRNA transfection with or without bafilomycin treatment overnight. Cells were treated with Dox for 48 hr before harvest. As a positive control, cells were treated with Na-pervanadate (PV, phosphatase inhibitor) for 30 min before harvest. Lysates were pulled down with anti-HA antibody or control IgG and subjected to Western blot analysis. Ratio, tyrosine phosphorylation of ZNRF3 normalized to total ZNRF3. Note that bafilomycin is enhancing pTyr-ZNRF3 levels but does not qualitatively affect the siPTPRK outcome.

Figure 3 C shows binding data between ZNRF3 and MET, before and after HGF addition to cells. There seems to be a differences between the amount of pMET input (bottom panel), total MET and MET brought down by ZNRF3-HA, comparing 15 and 30 minutes. Perhaps the authors can comment on this.

Upon HGF stimulation, MET is recognized by the E3 ligase CBL and is subjected to ubiquitin mediated proteasomal degradation (PMID: 12244174). This negative feedback likely causes decreasing MET levels. The fact that HGF induces binding of ZNRF3 to MET despite decreasing MET levels, emphasizes the significance of the HGF effect.

The figures are nicely done, with the diagrams. But the yellow zone in Figure 1G is not explained.

We have now removed the yellow zone.

Reviewer #2:1. All experiments revealing a functional link between MET and ZNRF3 were performed in a single cell line (H1703 cells), and findings in this cell line could not be reproduced in HEK293T cells (discussion). Also, a lack of functional/biological readouts prevents an interpretation of the biological relevance of the presented model. A demonstration that the presented mechanism is conserved and broadly relevant to other cells and model systems would strengthen this study.

We now tested also an alternative lung adenocarcinoma cell line, H3122, and inhibiting MET with Crizotinib reduced mature LRP6, an effect that was abolished when ZNRF3/RNF43 where siRNA-inhibited (Figure 5 —figure supplement 1A). This result supports that MET regulates WNT receptors via ZNRF3/RNF43 also in H3122 cells.

2. The authors state that all experiments were performed in the presence of Bafilomycin A1 (p5), which has many caveats, in particular when evaluating protein trafficking behavior. Can the authors exclude that their findings are a merely a curiosity of conditions in which endosomal acidification is prevented? Based on the literature, endosomal organization, endocytosis as well as recycling rates are affected by these conditions. To substantiate the proposed model, at least some of the functional experiments (e.g. HGF treatment-mediated alterations in β-catenin levels, ZNRF3 cell surface stabilization, FZD and LRP6 turnover assays) should be performed in absence of the drug.

We apologize for the misunderstanding regarding bafilomycin treatment (p5). Bafilomycin was only used in the few western blot analyses which analyzed pTyr-ZNRF3, because the tyrosine phosphorylation signal is very weak and labile. Bafilomycin was *not* added in any other experiments. The bafilomycin trick to boost pTyr-ZNRF3 was already employed in Chang et al., where we demonstrated qualitatively the same- but weaker effects without bafilomycin (see Author response image 1) . We now have modified the sentence.

3. For HGF and Wnt pathways, multiple points of crosstalk have been reported. To judge the relative contribution of HGF-MET regulation of ZNRF3 versus previously reported crosstalk events, the levels of HGF-induced β-catenin stabilization and β-catenin-dependent reporter activity should be evaluated in ZNRF3/RNF43-ko cells.

Previously described HGF effects on Wnt signalling act downstream of ligand-receptor signalling. However, in H1703 cells such downstream-acting mechanisms that bypass Wnts seem not to be involved since without Wnt3a HGF did not induce b-catenin accumulation (Figure 5D-E). Moreover, we carried out the experiment you suggested, albeit with siRNA instead of ZNRF3/RNF43-ko cells, to safe time: Wnt3a-induced β-catenin was enhanced two-fold by HGF treatment and siZNRF3/RNF43 blocked this effect (Figure 5 —figure supplement 1C). This result confirms that at least in in H1703 cells, HGF cross-talk with Wnt signalling is mostly by the here-described mechanism.

4. The authors make a comparison between the activities HGF and RSPO. Both factors mediate pathway activation by acting on ZRNF3, albeit by different mechanisms that involve either ZNRF3 stabilization at the plasma membrane (HGF) or an increase in ZNRF3 turnover (RSPO). The outcome of both pathways is Wnt receptor stabilization. To judge the relative contribution of HGF-mediated pathway stimulation, can the authors make a direct comparison of RSPO and HGF-mediated potentiation of Wnt-induced pathway activation, using reporter assays? Also, do these pathways act synergistically?

We now compared side-by-side HGF and RSPO in Wnt stimulation (Author response image 2) . Both RSPO1 and HGF enhanced Wnt3a induced β-catenin dose dependently and to about the same extend. However, the profiles were different. While RSPO1 induced a monotonic increase, the HGF effect decreased at higher doses, possibly because of too high MET internalization.

**Author response image 2. sa2fig2:** Comparison of recombinant human RSPO1 and HGF in Wnt activation. Immunoblot analysis of cytosolic β-catenin in H1703 cells upon 2h treatment of Wnt3a with different concentrations of recombinant human HGF or RSPO1. Ratio, relative levels of β-catenin normalized to GAPDH. Data show a representative result from two independent experiments with similar outcome.

We now also tested for HGF and RSPO synergism. We treated cells with a sub-optimal dose of Wnt3a and added HGF or RSPO3 alone or in combination and then monitored b-catenin accumulation. Interestingly, HGF and RSPO3 strongly synergized in inducing b-catenin accumulation (Figure 5 —figure supplement 1D). This observation suggests that HGF and RSPO act non-redundantly to target distinct ZNRF3 pools. This impressive synergism also suggests that HGF may be useful in growing Wnt-dependent organoids. Thank you for suggesting this experiment, which led to an interesting finding!

5. In line 234-236, the authors speculate that HGF inhibition might be beneficial for treatment of WNT-dependent cancers. However, as WNT-dependent colorectal cancers commonly display a lack of ZNRF3 expression, either due to mutation or gene silencing (e.g. Lannagan et al., Gut 2019), this poses a conceptual problem.

Not all CRCs lack ZNRF3 expression. For example, Bond et al. (PMID: 27661107) found no significant change on ZNRF3 expression between BRAF^V600E^ and matched normal mucosa samples. Moreover, 8% of CRCs show high expression of RSPO2/3, which is mutually exclusive with RNF43 mutation (Kleeman and Leedham 2020, PMID: 33202731), suggesting that in those CRCs hyperactivation of WNT pathway is achieved via RSPO driven ZNRF3/RNF43 sequestration. Finally, Storm et al. (Storm et al., 2016, PMID: 26700806) showed that PTPRK-RSPO3-fusion triggers CRC, and in all likelihood via ZNRF3/RNF43. Some of these CRCs may benefit from MET inhibition.